# Improving Nasal Protection for Preventing SARS-CoV-2 Infection

**DOI:** 10.3390/biomedicines10112966

**Published:** 2022-11-17

**Authors:** Riccardo Nocini, Brandon Michael Henry, Camilla Mattiuzzi, Giuseppe Lippi

**Affiliations:** 1Unit of Otorhinolaryngology, Department of Surgery, Dentistry, Paediatrics and Gynaecology, University of Verona, Piazzale L.A. Scuro 10, 37134 Verona, Italy; riccardo.nocini@aovr.veneto.it; 2Division of Nephrology and Hypertension, Cincinnati Children’s Hospital Medical Center, 3333 Burnet Ave, Cincinnati, OH 45229, USA; bmhenry55@gmail.com; 3Service of Clinical Governance, Provincial Agency for Social and Sanitary Services (APSS), Via Alcide Degasperi 79, 38123 Trento, Italy; camilla.mattiuzzi@apss.tn.it; 4Section of Clinical Biochemistry and School of Medicine, University of Verona, Piazzale L.A. Scuro 10, 37134 Verona, Italy

**Keywords:** COVID-19, SARS-CoV-2, nasal spray, infection

## Abstract

Airborne pathogens, including SARS-CoV-2, are mainly contracted within the airway pathways, especially in the nasal epithelia, where inhaled air is mostly filtered in resting conditions. Mucosal immunity developing after SARS-CoV-2 infection or vaccination in this part of the body represents one of the most efficient deterrents for preventing viral infection. Nonetheless, the complete lack of such protection in SARS-CoV-2 naïve or seronegative subjects, the limited capacity of neutralizing new and highly mutated lineages, along with the progressive waning of mucosal immunity over time, lead the way to considering alternative strategies for constructing new walls that could stop or entrap the virus at the nasal mucosa surface, which is the area primarily colonized by the new SARS-CoV-2 Omicron sublineages. Among various infection preventive strategies, those based on generating physical barriers within the nose, aimed at impeding host cell penetration (i.e., using compounds with mucoadhesive properties, which act by hindering, entrapping or adsorbing the virus), or those preventing the association of SARS-CoV-2 with its cellular receptors (i.e., administering anti-SARS-CoV-2 neutralizing antibodies or agents that inhibit priming or binding of the spike protein) could be considered appealing perspectives. Provided that these agents are proven safe, comfortable, and compatible with daily life, we suggest prioritizing their usage in subjects at enhanced risk of contagion, during high-risk activities, as well as in patients more likely to develop severe forms of SARS-CoV-2 infection.

## 1. Introduction

Coronavirus Disease 2019 (COVID-19) is a life-threatening, highly infectious pathology that was first diagnosed at the end of 2019 in the Chinese city of Wuhan [1]. The disease is caused by a beta coronavirus, which was called severe acute respiratory syndrome coronavirus 2 (SARS-CoV-2) by the Coronaviridae Study Group of the International Committee on Taxonomy of Viruses [2]. Since 2019, the disease has spread all over the world, was defined a pandemic by the World Health Organization (WHO) on March 2020 [3], and has since become the seventh most lethal infectious outbreak throughout the traceable human history [4], with 628 million official cases and 6.6 million attributable deaths reported in the WHO COVID-19 Dashboard at the time of writing this article (3 November 2022) [5].

## 2. Pathways of Host Cell Penetration by SARS-CoV-2

COVID-19 is a typical respiratory infectious disease. Although it was earlier assumed that SARS-CoV-2 could only be transmitted with droplets, it has then become evident that contagion could also be driven by aerosols [6], and to a lower extent through fomite [7,8]. The epithelial cells of conducting airways, especially those of the upper respiratory tract (nose, mouth, oropharynx), are hence the leading port of entrance of this airborne virus within the human host, followed by alveolar cells residing within the lung tissue [9].

The cell penetration of SARS-CoV-2 is mainly mediated by its spike protein, which binds effectively through its receptor-binding domain (RBD) to the human receptor angiotensin converting enzyme 2 (ACE2) upon proteolytic cleavage and activation by a number of human enzymes such as transmembrane serine protease 2 (TMPRSS2), furin, furin-like, trypsin-like and cathepsin proteases, among others [10]. Cell entry is also facilitated by a number of viral attaching factors such as heparan sulfate proteoglycans (HSPGs), phosphatidylserine (PS) receptor, neuropilin-1 (NRP-1), CD147 and C-type lectins, which act by enhancing virus adhesiveness to the host cell membrane [11]. This mechanism, which largely predominated during the initial phases of the pandemic corresponding to the spread of the ancestral SARS-CoV-2 strain and early variants such as alpha, beta and delta, has been joined by alternative pathways of cell penetration observed in the highly mutated Omicron variants which replaced previous strains. Briefly, reliable evidence has been provided that the Omicron sublineages, other than replicating prevalently in the upper respiratory tract [12], may also penetrate the host cell by endocytosis and sorting within endolysosomes, employing an acid-activated cathepsin L mechanism [13]. This evidence has been recently confirmed by an elegant study published by Iwata–Yoshikawa et al. [14], who showed that Omicron sublineages seem to use the furin/TMPRSS2-dependent entrance pathway less efficiently, thus penetrating the host cells prevalently through a cathepsin-dependent endocytosis pathway in TMPRSS2-expressing cells. Cell to cell transmission is a third important mechanism that has been elucidated for explaining host cell penetration by SARS-CoV-2, whereby endosomal membrane fusion [15] or generation of syncytia mediated by the binding of spike protein expressed on the surface of infected cells with ACE2 present in adjacent and uninfected cells [16] may allow the spread of the virus in the nearby respiratory tissue. A fourth mechanism of cell invasion involves SARS-CoV-2 bearing microparticles that may be present in the circulation of infected subjects as early as one day before symptoms onset, and which could persist for up to four weeks afterwards [17]. These SARS-CoV-2-positive extracellular vesicles, mostly generated by budding and fission of plasma membrane of SARS-CoV-2-infected cells, could be released and fuse with the membrane of other host cells, releasing the viral genome within the cytoplasm of uninfected cells.

Altogether, these four mechanisms (summarized in Figure 1) may hence synergically contribute to foster or amplify the spread of SARS-CoV-2 within the host.

## 3. The Role of Mucosal Protection in Preventing SARS-CoV-2 Infection

As previously noted, the epithelial cells of the upper respiratory tract, especially those of the nasal mucosa, are the main ports of entry for the current SARS-CoV-2 Omicron sublineages, a theory confirmed by the much higher prevalence of upper respiratory tract symptoms developing during acute Omicron infection compared to former variants [18,19]. Owing to this premise, it is easily understandable that the presence of an effective nasal “barrier” that could stop or entrap the virus at the mucosa surface would represent a highly effective means for preventing SARS-CoV-2 infection, since the nose filters thousands liters of air every day, conveying to the lungs over 90% of the total volume of inhaled air in resting conditions [20].

### 3.1. Natural or Vaccine-Elicited Mucosal Immunity

The normal human mucosa is already predisposed to develop an immunologic protection by means of generating neutralizing secretory antibodies (IgG and, especially, dimeric IgA) as a result of a response to natural infection or vaccination [21]. Convincing evidence has been provided that anti-SARS-CoV-2 specific IgA antibodies may provide an effective defense by inducing mucosal immunity within the respiratory system and thus lowering the risk of acute viral infection [22]. The real clinical benefits of mucosal (especially IgA-mediated) protection have been highlighted in several studies, such as that published by Hennings et al. [23], who showed that effective protection against the risk of contracting an acute SARS-CoV-2 infection seems to be principally associated with IgA-dominated anti-SARS-CoV-2 antibody response. In another recent work published by Havervall et al. [24], the authors showed that recipients of three doses of different types of COVID-19 vaccines with high titers (i.e., ≥75th percentile) of anti-SARS-CoV-2 spike mucosal IgA antibodies had 65% lower risk of developing SARS-CoV-2 infection compared to those with lower levels. A higher efficacy against SARS-CoV-2 infection has also been noted for high-titer mucosal IgA than for high-titer mucosal IgG (i.e., 65% vs. 27%). Unfortunately, however, there are at least three major drawbacks that plague immunologic protection in the mucosa of nose and other districts of the respiratory airways.

The foremost issue is that such protection would not efficiently work in COVID-19 seronegative or SARS-CoV-2 naïve individuals, i.e., those who have not been previously infected by whatever type of SARS-CoV-2 variant. Even the protective role of preexisting immunity developed after exposure to other coronaviruses remains highly controversial, though mounting evidence suggests that it may not work efficiently to prevent SARS-CoV-2 infection [25].

The second important caveat is that a former SARS-CoV-2 infection or a COVID-19 vaccination would not always provide sufficient protection when new and highly mutated lineages become prevalent. According to recent data published by Malato et al. [26], a previous infection with one of the pre-Omicron SARS-CoV-2 variants such as Wuhan-Hu-1 and Alpha would only provide modest protection (i.e., around 50%) against an infection with the Omicron BA.4/5 sublineages. Even former infection with the Omicron BA.1/2 sublineages would only confer a partial protection (i.e., around 75%) against new Omicron BA.4/5 infection [26]. Similar data were published by Altarawneh et al. [27], who also showed that a pre-Omicron acute infection was associated with only 33% protection against any new BA.4/5 infection.

Comparable considerations can be made for COVID-19 vaccination, whereby the administration of the former generation of “monovalent” vaccines (i.e., those only based on the prototype/ancestral SARS-CoV-2 strain), provides limited protection against infection by new Omicron sublineages. For example, the results of the recent meta-analysis published by Meggiolaro and colleagues [28] evidenced that although the efficacy of a booster COVID-19 monovalent vaccine dose remained as high as 86% against Omicron-related hospitalization, the protection against any type of Omicron infection was below 50%. Particular concern has then been raised by the fact that even the administration of the novel “bivalent” (i.e., ancestral-BA.4/5) COVID-19 vaccines may not work efficiently to prevent SARS-CoV-2 infection sustained by other emerging variants such as BA.2.75 and BA.2.75.1 [29].

The gradual waning of natural or vaccine-elicited immunity against SARS-CoV-2 is the third paradigmatic aspect that would contribute to limit the efficacy of mucosal immunity. Although we have previously seen that a previous SARS-CoV-2 infection or COVID-19 vaccination may be capable of triggering a robust mucosal immunity, the duration of such protection is limited over time. A study published by Isho and colleagues [30] showed that both anti-SARS-CoV-2 IgA and IgG neutralizing antibodies developed after an acute SARS-CoV-2 infection persist in saliva for no longer than 100–150 days. Similarly, Sano et al. [31] reported that both anti-SARS-CoV-2 IgA and IgG neutralizing antibodies elicited by COVID-19 vaccination gradually wane over time in SARS-CoV-2 seronegative individuals, with mucosal concentrations returning below measurable levels within 150–200 days. These results were replicated by Planas et al. [32], who also found that BA.5 neutralization in the nasal mucosa is low after vaccination with a monovalent mRNA-based COVID-19 vaccine, and considerably decreases after around 5–6 months even in subjects with BA.1/2 breakthrough infection.

### 3.2. Artificial Mucosal Protection within the Nose

Owing to the important limitations characterizing anti-SARS-CoV-2 mucosal immunity highlighted in the previous paragraphs, development and implementation of “artificial” means of mucosal protection within the nose may be seen as an intriguing and appealing strategy for reducing the risk of SARS-CoV-2 (re)infection. Besides the administration of virucidal agents [33,34], which are typically used for lowering the viral load in infected tissues and not for preventing an acute infection, or wearing face masks, whose role in lowering the risk of contagion sustained by the vast majority of respiratory pathogens is now virtually unquestionable [35,36] though being plagued by unfavorable biological and forensic implications [37], one emerging strategy encompasses the use of nasal sprays that function by impeding or disrupting the direct binding of SARS-CoV-2 to human respiratory epithelial cells [38]. The efficiency of some of these agents is confirmed by mounting scientific evidence that we aim to briefly summarize in the following parts of this article.

#### 3.2.1. In Vitro and Animal Studies

De Vries et al. designed lipopeptide fusion inhibitors with the purpose of inhibiting membrane fusion between SARS-CoV-2 and the host cell [39]. Daily intranasal administration of a dimeric SARS-CoV-2 highly conserved heptad repeat domain at the C terminus of the spike protein (HRC)-lipopeptide fusion inhibitor was found to completely abolish SARS-CoV-2 direct-contact transmission in ferrets. In a separate study, Shapira et al. identified a small-molecule compound (N-0385) which inhibits type-II transmembrane serine proteases [40], and found that it was highly efficient (in the nanomolar range) for preventing SARS-CoV-2 infection of Calu-3 cells. In a separate investigation, Lu et al. administered a cocktail of anti-SARS-CoV-2 broadly neutralizing antibodies (F61/H121; 20 mg/kg body weight) to K18-hACE2 mice [41], reporting that SARS-CoV-2 RNA could not be detected in mice tissue after challenge with the Omicron variant.

Yip et al. tested the properties of a commercially available Chinese medicine nasal spray called Allergic Rhinitis Nose Drops (ARND) and containing 11 Chinese medicinal herbs [42]. In a model based on SARS-CoV-2 pseudovirus of upper respiratory tract epithelial A549 cells over-expressing ACE2, pre-treatment with ARND was effective to reduce >50% pseudovirus infection.

Fais et al. explored the efficacy of AM-301 nasal spray (containing water, propylene glycol, mono-, di- and triglycerides, mannitol, magnesium aluminum silicate, xanthan gum, disodium EDTA and citric acid) against SARS-CoV-2 infection on a 3D-model of primary human nasal airway epithelium [43]. Administration of this agent 24 h before contact with SARS-CoV-2 efficiently reduced cell infection by 12-fold.

The clinical efficacy of a nasal spray containing 1% astodrimer sodium was tested by Paull et al. through intranasal administration in a K18-hACE2 mouse model [44]. The administration of this compound 60 min before SARS-CoV-2 challenge almost completely abolished viral infection.

Bentley and Stanton investigated the effectiveness of three different hydroxypropyl methylcellulose (HPMC)-based nasal sprays (93% HPMC, 2% peppermint powder and 5% European wild garlic powder; 95% HPMC, 2% peppermint powder and 3% allicin powder; 98.5% HPMC and 1.5% peppermint powder) to prevent in vitro SARS-CoV-2 infection [45]. Pre-treatment of ACE2-expressing VeroE6 cells with each of these compounds at a concentration of 6.4 mg/3.5 cm^2^ was effective to prevent SARS-CoV-2 infection, but was also associated with almost total inhibition of virus release from infected cells.

Bovard and colleagues tested a nebulized iota-carrageenan (IC) nasal spray in a 3-D model of reconstituted nasal epithelium [46]. At 48 and 72 h post-treatment with 7.2–21.8 μg/cm^2^ IC, molecular testing reveled substantially lower intracellular SARS-CoV-2 RNA (between 3–5 orders of magnitude). In a separate investigation, Morokutti–Kurz et al. also demonstrated that 30 min pre-treatment with 10 μg/mL IC can efficiently inhibit by over five-fold the infection of Vero B4 cells with SARS-CoV-2 Spike Pseudotyped Lentivirus [47]. The efficacy of a polysaccharide-based spray containing a mixture of gellan and λ-carrageenan was also tested by Moakes et al. [48]. The 48-h incubation of Vero cells with 1% concentration of this compound was highly effective to prevent SARS-CoV-2 infection, resulting in almost complete inhibition of viral entry.

Pyrć et al., investigated the effect of a positively charged polymer containing N-palmitoyl-N-monomethyl-N,N-dimethyl-N,N,N-trimethyl-6-O-glycolchitosan (GCPQ) in a model of human airway epithelial cells [49]. The authors found that SARS-CoV-2 infection could be efficiently prevented using a final concentration of 500 μg/mL of this compound. Moreover, intranasal administration of 20 mg/kg GCPQ was effective to prevent viral colonization of respiratory tract and brain in ACE2-expressing transgenic mice.

Zaderer et al., investigated the effect of ColdZyme mouth spray (composed of water, menthol, glycerol, calcium chloride and trypsin) administration to fully differentiated, polarized human epithelium cultured cells [50]. Using SARS-CoV-2 Omicron BA.1 and BA.4/5 sublineages, the authors found that this agent efficiently inhibited respiratory tissue infection, also preventing intracellular complement activation, inflammation and impairment of trans-epithelial integrity. In a subsequent investigation, the same team of authors replicated these findings in a model of human bronchial epithelial cells [51], where the use of the compound was effective to inhibit SARS-CoV-2 cell binding and infection, while concomitantly preventing complement activation and cell injury.

#### 3.2.2. Human Clinical Studies

Evidence emerged by the study of Figueroa and colleagues, using an IC-containing nasal spray, which was randomly administered with placebo to 394 Hospital workers [52]. All study subjects were asked to self-administer 1 puff (0.10 mL, 0.17 mg of IC or placebo) four times daily in both nostrils. The number of diagnoses of SARS-CoV-2 infection after 21 days of treatment was found to be significantly lower in subjects who used the IC nasal spray (2/196) compared to those who used the placebo (10/198; absolute risk reduction, 4%; 95%CI, 0.6–7.4%). No side-effects analysis was carried out in this investigation.

Paolacci et al. administered a nasal spray containing α-cyclodextrin and hydroxytyrosol to 149 healthy volunteers at high risk of SARS-CoV-2 infection due to their occupation [53]. During 30 days of follow-up, none of these subjects was diagnosed with COVID-19, although no SARS-CoV-2 infection also occurred in 76 control individuals who did not use the nasal spray during the same period. No side-effects analysis was carried out in this work.

Preliminary but interesting evidence has also been provided that an anti-SARS-CoV-2 neutralizing monoclonal antibodies (mAbs)-based nasal spray may provide good protection against SARS-CoV-2 infection. In a small-scale clinical trial, Lin et al. explored the efficacy of a single nasal spray containing the anti-SARS-CoV-2 35B5 mAb, which was tested in 30 healthy volunteers [54]. It was found that the mAB 35B5 concentration in a nasal mucosal specimens remained significant up to 72 h after spray administration (1 mg/mL 35B5 mAb diluted in 50% Dulbecco’s phosphate-buffered saline with 50% glycerol), concomitantly conveying up to 24–48 h efficient in vitro neutralization of several SARS-CoV-2 variants of concern (VOCs), including Omicron.

Recent evidence has been provided that the anticoagulant drug heparin displays pleiotropic antiviral properties, mostly by binding to the spike protein of SARS-CoV-2 and thus inhibiting host cell infection [55]. It is hence not surprising that its administration through the unconventional nasal route has been conceived as a prophylactic treatment against SARS-CoV-2 infection [56]. To this end, Eder et al., carried out a single-center, open-label intervention study aimed at exploring the effect of low molecular weight heparin (LMWH) inhalation for preventing SARS-CoV-2 infection [57]. By means of a nebulizer, 33 subjects received 4500 IU Enoxaparin or placebo in the right or left nostrils, respectively. Nasal epithelial cells were then collected by brushing and exposed to both authentic SARS-CoV-2 isolate and SARS-CoV-2 pseudovirus. In both cases binding to, and infection of, human nasal cells were found to be substantially inhibited by enoxaparin for up to 16 h.

The study protocol of a phase III, double-blind, randomized, single-centre clinical trial for establishing the efficacy of carrageenan-based nasal spray (purified water, 0.5% sodium chloride, 1.2 mg iota-carrageenan and 0.4 mg kappa-carrageenan) for lowering the risk of SARS-CoV-2 infection was presented in September 2022 [58], but results are still unavailable.

## 4. Conclusions

Although widespread COVID-19 vaccination and gradual mitigation of SARS-CoV-2 pathogenicity have both synergically contributed to considerably attenuate the clinical impact of COVID-19 over time, this infectious disease is still causing a huge number of mild symptomatic infections (which may hence cause substantial social and economic consequences due to the risk of transmitting the infection, or to the need of quarantine and isolation), and continues to generate a significant burden of hospitalizations, especially those attributable to acute infection of unvaccinated and/or fragile individuals [59]. The identification and implementation of reliable and practical strategies for preventing inter-human transmission should hence be seen as a top priority to enable effective management of this ongoing pandemic.

Airborne pathogens are mainly contracted within the respiratory pathways, especially through the nose, since airborne particles are mostly filtered within the nasal airway [60]. The important role played by the nasal epithelia in COVID-19 has been further magnified after emergence of the many Omicron sublineages, which exhibit highly efficient replication within the nasal tissue [61]. Mucosal immunity in this district, either developing after acute SARS-CoV-2 infection or COVID-19 vaccination, may hence represent one of the most efficient deterrents for preventing direct contact between the virus and the host cells. Nonetheless, the lack of such efficient protection in SARS-CoV-2 naïve or seronegative subjects, the limited capacity of neutralizing new and highly mutated viral lineages, along with the progressive waning of mucosal immunity over time, lead the way to adopting alternative strategies aimed at building new walls that could efficiently stop the virus at the mucosal surface, especially that of the nasal district, which is the area prevalently colonized by the new Omicron sublineages [62,63]. This is especially true considering that preliminary trials with intranasal COVID-19 vaccines aimed at preventing SARS-CoV-2 infection at the point of viral entry have generated disappointing results in terms of effective protection so far, mostly attributed to failed generation of an efficient mucosal antibody response [64].

Among the various infection prevention strategies, those based on generation of physical barriers over nasal epithelial (i.e., using compounds displaying mucoadhesive properties, which act through hindering, entrapping or adsorbing the virus on the sprayed layer), or those hampering the association of the virus with its cellular receptors (i.e., administering anti-SARS-CoV-2 neutralizing antibodies or agents that inhibit priming or binding of SARS-CoV-2 spike protein) are truly appealing perspectives. A vast array of compounds have been found effective for preventing SARS-CoV-2 infection in vitro or in animal models to date, whilst the number of sizeable randomized clinical trials is still very limited in humans. This evidence paves the way to planning comprehensive clinical trials, specifically aimed at testing, as well as comparing safety and effectiveness of these intranasally administered agents for preventing SARS-CoV-2 infection in humans before recommending their widespread administration.

Nonetheless, provided that these agents will prove to be safe, comfortable, and compatible with daily life, we suggest that their usage should be prioritized in subjects at higher risk of contagion, during high-risk activities, as well as in patients at enhanced risk of developing severe COVID-19 illness (Table 1).

To this end, it is noteworthy that the German Society of Hospital Hygiene (DGKH) has recently endorsed the use of nasal sprays based on Carragelose, 3 times/daily in older people residing in elderly-care facilities or rehabilitation facilities and sharing communal activities, at family gatherings or professional meetings, schools, kindergartens and religious occasions [65]. A clinical trial aimed to test the efficiency of nasal filters for preventing airborne contagion by SARS-CoV-2 is also underway [66]; these devices may have main application in high-risk conditions, such as in healthcare setting where COVID-19 patients are cured.

## Figures and Tables

**Figure 1 biomedicines-10-02966-f001:**
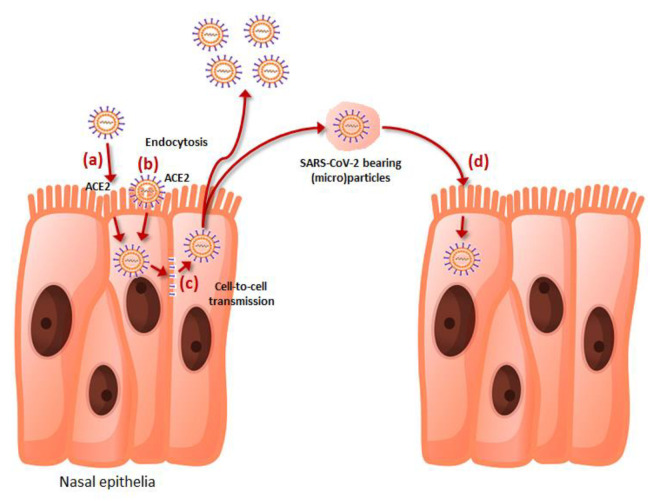
The pathways of epithelial cell penetration by SARS-CoV-2, encompassing (**a**) direct attachment to angiotensin converting enzyme 2 (ACE2) on host cell surface, (**b**) cathepsin-dependent endocytosis; (**c**) cell-to-cell transmission, and (**d**) fusion of SARS-CoV-2-bearing extracellular particles with host cell membrane.

**Table 1 biomedicines-10-02966-t001:** The preferential usage of nasal sprays for preventing SARS-CoV-2 infection.

Categories of subjects at higher risk of contagion ○ Unvaccinated and/or seronegative subjects ○ Healthcare staff (especially those caring SARS-CoV-2 positive patients) ○ Family members living with or assisting SARS-CoV-2 positive relatives High-risk activities ○ Indoor mass gatherings ○ Eating meals ○ Staying for >15 min in crowded and poorly ventilated environments without physical protections (e.g., basically without waring facemasks) Subjects at enhanced risk of developing severe disease ○ Older subjects ○ Immuncompromised patients ○ Patients with important comorbidities

## Data Availability

Not applicable.

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
