# Peer review of "Improving Nasal Protection for Preventing SARS-CoV-2 Infection"

_biomedicines, 2022, doi:10.3390/biomedicines10112966_

Round 1

Reviewer 1 Report

            The authors propose a Perspective on nasal protection for preventing SARS-CoV-2 infection. The information is interesting and pertinent. Some concernss are included that could help to improve the manuscript

1.      Introduction, page 2, ref 14 shows exactly the contrary that the authors mention: mice knockout for TMPRSS2 gene are less susceptible to the Omicron sublineages, suggesting that in vivo this variant still uses this serin protease for infection.

2.      Figure 1: in pathway b, the virus also uses ACE2 as receptor.

Author Response

The authors propose a Perspective on nasal protection for preventing SARS-CoV-2 infection. The information is interesting and pertinent. Some concernss are included that could help to improve the manuscript

  • We are thankful to the referee for the globally favourable comments on our manuscript. We’ll do our best to improve it according to the referee’s suggestions.

  1. Introduction, page 2, ref 14 shows exactly the contrary that the authors mention: mice knockout for TMPRSS2 gene are less susceptible to the Omicron sublineages, suggesting that in vivo this variant still uses this serin protease for infection.
  • ANSWER: Good point, thanks! Text revised accordingly, adjusting the original text of that paper [Ref. 14] that states: “Here, we confirm that Omicron uses the furin/TMPRSS2-dependent pathway inefficiently and enters cells mainly using the cathepsin-dependent endocytosis pathway in TMPRSS2-expressing VeroE6/TMPRSS2 and Calu-3 cells” [See: https://www.nature.com/articles/s41467-022-33911-8].

  1. Figure 1: in pathway b, the virus also uses ACE2 as receptor.
  • ANSWER: Good point, thanks! Figure 1 revised accordingly.

Reviewer 2 Report

Biomedicines-2047842 

This is a perspective that  addresses an important topic. However, the presentation of the information needs additional work. Specific comments are as follows:

The table 1 is unnecessary. The authors should just list the limitations in the paragraph before they describe of each of them.

The in vitro and animal studies section is very descriptive and in the last paragraph is repetitive e.g. "In a subsequent investigation, Zaderer...In a subsequent investigation, the same team..."

Page 7, the mention of Table 1 in the last paragraph is irrelevant.  

Author Response

This is a perspective that addresses an important topic. However, the presentation of the information needs additional work. Specific comments are as follows:

  • We are thankful to the referee for the globally favourable comments on our manuscript. We’ll do our best to improve it according to the referee’s suggestions.

The table 1 is unnecessary. The authors should just list the limitations in the paragraph before they describe of each of them.

  • ANSWER: Thanks for this suggestion. We agree. Table 1 has been removed and information only provided in the text, as follows: “Nonetheless, the lack of such efficient protection in SARS-CoV-2 naïve or seronegative subjects, the limited capacity of neutralizing new and highly mutated viral lineages, along with progressive waning of mucosal immunity over time…”.

The in vitro and animal studies section is very descriptive and in the last paragraph is repetitive e.g. "In a subsequent investigation, Zaderer...In a subsequent investigation, the same team..."

  • ANSWER: Thanks for raising this issue. We have deleted all repeated phrasing and the manuscript is indeed more readable now.

Page 7, the mention of Table 1 in the last paragraph is irrelevant. 

  • ANSWER: Thanks for this suggestion. We agree. Table 1 has been removed and information only provided in the text, as follows: “Nonetheless, the lack of such efficient protection in SARS-CoV-2 naïve or seronegative subjects, the limited capacity of neutralizing new and highly mutated viral lineages, along with progressive waning of mucosal immunity over time…”.